# Targeting Mitochondria and Metabolism in Acute Kidney Injury

**DOI:** 10.3390/jcm10173991

**Published:** 2021-09-03

**Authors:** Ying Li, Mark Hepokoski, Wanjun Gu, Tatum Simonson, Prabhleen Singh

**Affiliations:** 1Division of Nephrology and Hypertension, University of California San Diego, San Diego, CA 92093, USA; yil026@health.ucsd.edu; 2VA San Diego Healthcare System, San Diego, CA 92161, USA; mhepokoski@health.ucsd.edu; 3Division of Pulmonary, Critical Care and Sleep Medicine, University of California San Diego, San Diego, CA 92093, USA; wagu@health.ucsd.edu (W.G.); tsimonson@health.ucsd.edu (T.S.)

**Keywords:** energy metabolism, kidney injury, mitochondria

## Abstract

Acute kidney injury (AKI) significantly contributes to morbidity and mortality in critically ill patients. AKI is also an independent risk factor for the development and progression of chronic kidney disease. Effective therapeutic strategies for AKI are limited, but emerging evidence indicates a prominent role of mitochondrial dysfunction and altered tubular metabolism in the pathogenesis of AKI. Therefore, a comprehensive, mechanistic understanding of mitochondrial function and renal metabolism in AKI may lead to the development of novel therapies in AKI. In this review, we provide an overview of current state of research on the role of mitochondria and tubular metabolism in AKI from both pre-clinical and clinical studies. We also highlight current therapeutic strategies which target mitochondrial function and metabolic pathways for the treatment of AKI.

## 1. Introduction

Acute kidney injury (AKI) is the abrupt decline in kidney function. Several definitions have been used for AKI in the past. The most current and preferred consensus definition has been provided by the Kidney Disease: Improving Global Outcomes (KDIGO) and is as outlined in Table 1. KDIGO has also provided staging of AKI (Table 1). There are several etiologies of AKI which fall into the categories of pre-renal (reduced renal perfusion without any tubular injury), intrinsic or acute tubular necrosis (due to direct tubular injury) or post-renal (obstruction). Pre-renal and intrinsic AKI are most common in hospitlalized patients. In this review, we focus on mechanisms of intrinsic AKI with tubular injury, which is caused by three common etiologies including renal ischemia, nephrotoxins, and sepsis.

AKI is a common complication that occurs in about 35% of all hospitalized patients and 50–70% of critically ill patients in the intensive care unit (ICU) [1,2,3,4,5]. AKI most often develops as part of a multiple organ failure syndrome and the mortality rate of patients with AKI in the ICU is 30–50%. In addition to increasing the risk of death, AKI is also an independent risk factor for progression to chronic kidney disease (CKD) and end-stage renal disease [6,7,8]. Despite this unacceptably high morbidity and mortality, effective strategies to prevent or reverse AKI are extremely limited due to its multifactorial pathogenesis. Sepsis-associated acute kidney injury, which accounts for up to 50% of all AKI with more than 70% mortality rate, is particularly challenging with complex pathophysiology [9]. Therapeutic strategies for AKI have been generic, supportive, and largely ineffective.

Mitochondrial dysfunction has recently emerged as one pathogenic mechanism that seems to be universal among various etiologies of AKI. The kidney is one of the most metabolic organs due to the high energy demands to support solute transport, particularly, tubular sodium reabsorption [10,11,12]. Under physiological conditions, mitochondrial oxidative phosphorylation (OXPHOS) is the predominant source of ATP generation in the kidney. The proximal tubule in the cortex and thick ascending limb in the outer medulla are rich in mitochondria and are the most metabolically active segments in the kidney. These segments are also more susceptible to injury in AKI. Pre-clinical and clinical studies have strongly implicated mitochondrial dysfunction as a critical player in the pathophysiology of AKI [13].

Cellular metabolism is comprised of various pathways to sustain cellular homeostasis including metabolic pathways focused on energy generation. The high metabolic demands of the kidney place renal energy metabolism central to cellular health and function. Energy metabolism under various physiological conditions has been described in the kidney, but until recently, there have been limited studies exploring it in pathophysiological conditions. Recent publications have highlighted metabolic reprogramming in various forms of CKD [14,15,16,17]. The role of altered energy metabolism in AKI is also increasingly being recognized.

In this review, we present our current knowledge regarding mitochondrial function and tubular energy metabolism in AKI. We also summarize the current evidence linking mitochondrial dysfunction and metabolic reprogramming with the development and progression of AKI. Finally, we discuss potential therapeutic approaches targeting mitochondrial dysfunction and tubular metabolism.

## 2. Overview of Renal Energy Metabolism

Energy metabolism varies among different components of the kidney as well as between metabolically active epithelial cells in different tubular segments in the kidney. This heterogeneity includes the capacity to utilize different metabolic pathways to generate energy and preferences in substrates to fuel the metabolic pathways. A detailed overview of renal metabolism is available [12]. Pertinent to this review, overall, the kidney derives most of its energy in the form of ATP from mitochondrial OXPHOS.

Mitochondria support the changing energy demands in cells via ATP generation by oxidative metabolism. They also play an important role in the regulation of a variety of cellular functions and processes, including generation and detoxification of reactive oxygen species, cellular stress adaptation, survival, and death [18]. The kidney is only second to the heart in terms of mitochondrial abundance per gram tissue mainly concentrated in the tubular cells, particularly in the proximal tubules and thick ascending limbs. OXPHOS is carried out within the cristae which are the folds in the inner mitochondrial membrane. Cristae increases the surface area for embedding the abundant proteins responsible for OXPHOS, ATP synthesis, proton gradient generation, and mitochondrial dynamics. The region enclosed by the inner mitochondrial membrane is the mitochondrial matrix, which is the site of tricarboxylic acid cycle (TCA) or Kreb cycle metabolism. Carbohydrates (glucose), lipids (fatty acids), and proteins (amino acids) are metabolized outside the mitochondria to produce intermediates such as acetyl CoA, which then enter the TCA cycle [19]. Oxidation of acetyl CoA provides NADH and FADH_2_ which are electron carrier molecules and transfer electrons via a chain of enzyme complexes (complex I to IV) known as the electron transport chain (ETC). Electron transport within the ETC generates a proton gradient across the inner membrane. The buildup of protons within the intermembrane space establishes an electrochemical gradient across the inner membrane, which provides the power to ultimately produce ATP via activation of ATP synthase (complex V). ATP generated in the mitochondria is transported out of the mitochondrial matrix by the adenine nucleotide translocases and ADP/ATP carrier proteins [20].

Proximal tubules are largely dependent on OXPHOS for ATP generation and utilize fatty acids, amino acids and lactate to provide metabolites (acetyl CoA) for the TCA cycle within the mitochondria [19]. They have limited capacity to catabolize glucose as a metabolic fuel under physiological conditions, despite handling high glucose fluxes due to its luminal reabsorption via sodium-glucose linked transporters and basal transport via facilitative glucose transporters [21]. Catabolism of glucose via glycolysis involves a series of enzymatic reactions which results in generation of pyruvate in the cytosol. It is important to note that pyruvate can then be shunted to the mitochondria *in aerobic conditions* to be utilized as a substrate for the TCA cycle and generate ATP via OXPHOS, whereas, *under anerobic conditions*, pyruvate is converted to lactate. Other parts of the kidney, including thick ascending limb, glomerular, and vascular cells can effectively utilize glucose as a preferred metabolic substrate via *aerobic* glycolysis [19,21]. Inner medulla, which exists under conditions of physiological hypoxia, relies on *anerobic* glucose metabolism. In terms of ATP generation, glucose oxidation via glycolysis and subsequent OXPHOS generates ~36 moles of ATP per glucose molecule *in aerobic conditions*, compared to 2 moles of ATP per glucose molecule in *anaerobic conditions* [12]. Utilization of other fuels for OXPHOS generates close to 100 moles of ATP per molecule of substrate [22]. Compared with other substrates, fatty acid oxidation and the TCA cycle yield the most ATP per gram. However, these processes also require more oxygen than utilization of other substrates such as lactate, glutamine, or glucose. Hence, fatty acid oxidation is the preferred substrate when availability of oxygen is not limited, such as in the cortex of healthy kidneys which includes proximal tubular cells, thick ascending limb, and distal convoluted tubules [12,19,21,23].

Other metabolic pathways such as gluconeogenesis are also site-specific. While proximal tubules have limited capacity to metabolize glucose via glycolysis under physiological conditions, they have a high capacity to synthesize glucose from precursors such as lactate, glutamate, and glycerol via gluconeogenesis [21,24,25]. Gluconeogenic enzymes are found almost exclusively in the proximal tubule [26]. The other nephron segments, from medullary ascending limb to medullary collecting tubule, have a high capacity for glucose oxidation as a source of ATP generation via glycolysis, but have no gluconeogenic potential [26]. Additional details regarding the fuel preferences and metabolic pathways along the nephron have been described before [12,19,23,26,27]. To summarize, most components of the kidney, except the inner medulla, utilize mitochondrial OXPHOS for ATP generation, regardless of fuel preferences (glucose, fatty acids, lactate, or amino acids).

## 3. Mitochondrial Dysfunction and Metabolic Derangements in AKI

### 3.1. Mitochondrial Morphology and Biogenesis

The role of mitochondrial dysfunction in the pathophysiology of AKI has been well-established. Alterations in mitochondrial morphology develop early after AKI. In ischemic and nephrotoxic AKI, several structural changes including decreased mitochondrial mass, disruption of cristae, mitochondrial swelling, and fragmentation have been described [28,29,30]. The role of mitochondrial dynamics has been elucidated in AKI. Increased expression of dynamin related peptide 1 (Drp1), which induces mitochondrial fission, was shown in ischemic and glycerol induced AKI [31]. In an elegant study, Brooks et al. described the disruption of mitochondrial dynamics and its pathogenic role in ischemic and nephrotoxic AKI [28]. In fact, mitochondrial fragmentation was shown to precede tubular cell injury and death, which leads to the release of cytochrome C and other mediators of apoptotic cell death. Pharmacological inhibition of fragmentation ameliorated AKI of various etiologies [28,31]. Other studies have shown altered mitochondrial dynamics at both the outer and inner mitochondrial membranes in AKI, indicating the important role of disruption of mitochondrial dynamics in AKI [32,33,34,35].Electron microscopy demonstrates an elongated mitochondria shape in the proximal tubules of healthy individuals. In kidney tissues of septic patients, enlarged swollen mitochondria with disrupted cristae in the proximal tubules have been reported without overt tubular necrosis and swollen/fragmented mitochondria have also been observed in renal tubules in patients exposed to ischemia-reperfusion (IR) injury [36,37].

Studies have elucidated the role of mitochondrial biogenesis in recovery from ischemic AKI using stimulators of peroxisome proliferator-activated receptor γ coactivator-1α (PGC1α), a master regulator of mitochondrial biogenesis [31,38,39]. Stimulation of PGC1α correlated with expression of ETC proteins and with rates of coupled and uncoupled respiration. In sepsis associated AKI (s-AKI), decreased expression of PGC1α has been demonstrated in different models- lipopolysaccharide infusion (LPS) and cecal ligation and puncture (CLP) [40]. However, it is not clear whether this a cause or consequence of s-AKI. A decrease in PGC1α during sepsis is insufficient by itself to cause renal injury, but an increase appears to be important in recovery from sepsis. Induction of PGC-1α and consequent mitochondrial biogenesis during recovery from sepsis has been observed in other organs as well. Decreased expression of PGC-1α has been observed in patients with AKI [41]. In renal transplant patients with delayed graft function, decreased PGC-1α expression correlated with incomplete recovery of graft function [42].

### 3.2. Mitochondrial Functional Alterations

In terms of functional changes, increases in mitochondrial NADH, dissipation of mitochondrial membrane potential, and complex I dysfunction in proximal tubules have been demonstrated in ischemic AKI [30,43]. In ischemic and myoglobulinuric AKI, reduction in various ETC proteins in the proximal tubule was shown [31]. Other studies have shown reduced expression of several key mitochondrial proteins and mitochondrial DNA content in AKI [31,44]. Reduced ATP production and decreased respiratory complex expression have also been reported in several AKI models [40,45,46,47,48,49]. Generation of free radicals during transfer of high energy electrons along the ETC in mitochondria has also been implicated in the pathophysiology of AKI. Oxidative stress from either increased reactive oxygen species (ROS) generation or reduced antioxidant expression or activity has been shown to result in activation of cellular injury and death pathways [47,50,51,52]. Cisplatin nephrotoxicity is associated with a decrease in cytochrome C oxidase activity and protein expression in the proximal tubule along with a decrease in antioxidant enzymes [47]. Decreases in cortical and outer medullary cytochrome C activity along with swollen mitochondria with rarified and disrupted cristae occurred at 18 h after LPS and 24 h after CLP [40]. Reduced mRNA levels of some ETC proteins were also seen at 18 h after LPS. Patil et al. measured respiratory complex activity in aged CLP mice (40 weeks old) and observed reduced activity of ETC complexes and low ATP levels at 36 h after CLP [49].

Recently, we have comprehensively described the temporal course of mitochondrial function in s-AKI in CLP mice [53]. Proximal tubules demonstrate evolving changes in mitochondrial function and bioenergetics. In the early phase at four hours after CLP, increases in mitochondrial content and biogenesis were observed in CLP kidneys. However, basal and ATP-linked mitochondrial oxygen consumption rates in proximal tubules were decreased in CLP kidneys. In the late phase, at 24 h after CLP, mitochondrial content along with markers of biogenesis and fusion were significantly reduced in kidneys. Remarkably, proximal tubular mitochondria displayed high reserve bioenergetic capacity, which enables cells to withstand cellular stress and increase in energy requirements. They also demonstrated increased maximal respiratory capacity and respiratory control ratio, which indicates high capacity for substrate oxidation and respiration with low proton leak in mitochondria. However, ATP levels were reduced in the CLP kidneys, presumably from low ATP demand or from decreased efficiency of ATP synthesis due to changes in mitochondrial morphology and dynamics. Another possibility is that the low ATP generation is a survival adaptation as a form of hibernation to endure cellular stress from sepsis. This hypothesis is supported by the paucity of cell necrosis observed in septic AKI.

Mitochondrial dysfunction promotes the release of mitochondrial damage-associated molecular patterns (mtDAMPs), which further amplify inflammation and injury and trigger apoptosis [54]. These mtDAMPs include mitochondrial DNA (mtDNA), which, when released into the cytoplasm, stimulates inflammasomes and downstream activation of inflammatory cytokine cascades [55]. Release of mtDAMPs in the circulation has been shown to activate innate immune pathways similar to those in sepsis, inducing systemic inflammatory response syndrome [56]. Plasma mtDAMPs levels are increased after trauma [54] and sepsis [57] and the circulating concentration of mtDNA correlates with mortality in critically ill patients with ARDS [58]. We have recently shown that renal IR results in mitochondrial dysfunction in remote organs and cell types, including the lung [59]. This correlated with an increase in the mtDAMPs and mtDNA in the plasma and bronchoalveolar lavage fluid. Finally, through metabolomic analyses, we provided evidence for a decrease in metabolites generation through lung mitochondrial metabolism due to renal mtDAMPs. A summary of mitochondrial structural and functional changes is shown in Figure 1.

### 3.3. Metabolic Derangements

Proximal tubules use various substrates (fatty acids, amino acids and lactate) as fuel for ATP generation via OXPHOS. As discussed above, under normal oxygen conditions, glucose is converted to pyruvate, which enters the mitochondrial TCA cycle. In hypoxia, pyruvate is converted to lactate. When this occurs even under aerobic conditions, it is referred to as the “Warburg” effect or aerobic glycolysis and has been described in proliferating cells [60,61]. Pyruvate dehydrogenase (PDH) converts pyruvate to acetyl CoA, committing it to OXPHOS [62]. Pyruvate dehydrogenase kinase (PDK) inhibits PDH by phosphorylation and curbs the pyruvate entry into OXPHOS. Thus, the PDH-PDK axis serves as a gatekeeper for energy metabolism [62,63]. There is accumulating evidence on the role of the PDH-PDK pathway in the pathogenesis of diabetes, heart failure, and cancer [63].

Glycolysis supports macromolecular synthesis (nucleotides, amino acids) needed for cell regeneration [60,61]. Glycolytic metabolism plays an important role in nephron development. Nephron progenitors exhibit high rates of glycolysis to support cell proliferation and self-renewal [64]. A recent study demonstrated metabolic switching during development in mice, where nephron progenitors at embryonic day 13.5 show increased glycolysis, mitochondrial respiratory capacity and ATP production compared with postnatal day zero [64]. Inhibition of glycolysis stimulated cell differentiation and nephrogenesis. Tubular metabolic reprogramming with increased glycolysis and reduced fatty acid oxidation has been described in polycystic kidneys [16,17], diabetic kidneys [14] and in transgenic models of kidney fibrosis [15].

There is evidence for metabolic reprogramming in AKI as well. In ischemic and cisplatin-induced AKI, impaired mitochondrial and peroxisomal fatty acid oxidation (FAO) and accumulation of non-esterified, free FA have been reported along with downregulation of FAO enzymes [46,65,66]. These free FA causes an energetic defect by mitochondrial de-energization leading to ATP depletion, uncoupling of oxidative phosphorylation and impaired recovery of mitochondrial membrane potential [46]. Within the first 24 h after ischemia, mRNA levels of glycolytic enzymes were upregulated while those of gluconeogenic enzymes were downregulated in the kidney [67]. However, in a recent study, no significant changes in protein levels of glycolytic enzymes were seen within 24 h after ischemia, other than the transient increase in pyruvate kinase liver/RBC isoform levels, expressed in normal proximal tubules, at 8 h with subsequent decrease [68]. Starting at two days after ischemia, increased protein levels of glycolytic enzymes were observed at different time points, with the most pronounced effect observed at day 14. At the day 14 time point, small clusters of atrophic tubules with surrounding fibrosis were identified within mostly recovered tubular epithelium. Increased expression of glycolytic enzymes by immunohistochemistry was seen in all proximal tubules, particularly prominent in the atrophic tubules. In another recent study, protein levels of pyruvate kinase M were increased at 24 h after severe IR in mice, but no change in hexokinase 1 was seen [69]. Interestingly, in this study, there was evidence for decreased renal gluconeogenesis after IR. Protein and mRNA levels of gluconeogenic enzymes were decreased at 24 h after IR. Serum levels of glucose were decreased, whereas lactate levels were increased at 48 h. Focused evaluation of glucose and lactate metabolism revealed reduced lactate conversion to glucose and reduced renal glucose production, consistent with impaired gluconeogenesis.

In other models of AKI, some metabolic alterations have been identified but the data are variable. Smith et al. demontrated elevated hexokinase activity in LPS-induced AKI in mice [70]. In another recent study, one day after CLP in mice, decreased glucose along with increased phosphoenol pyruvate, pyruvate, and lactate levels were increased supporting increased glycolysis [71]. We recently published our observations at 24 h after CLP in mice [53]. There was decreased expression of FAO enzymes-carnitine palmitoyltransferase I and II and acyl-CoA oxidase 2. There was no change in the expression of glycolytic enzymes earlier in the pathway—hexokinase 1 and 2; however, there was an increase in the mRNA and protein expression of rate-limiting glycolytic enzyme, phosphofructokinase, which commits glucose to glycolysis. Pyruvate kinase M is another rate-limiting enzyme which catalyzes the final step in glycolysis, and its expression was also increased in the CLP kidneys. We also observed decreased protein expression of PDH, which catalyzes the conversion of pyruvate to acetyl-CoA to enter TCA cycle in the mitochondria. Consistent with this, the expression of the inactive form of PDH, phospho-PDH, was increased significantly in CLP kidneys. These data show upregulation of glycolytic pathway in the CLP kidney. However, assessment of functional glycolysis by measuring extracellular acidification rate as well as proton extrusion rate revealed that a reduction in all parameters of glycolysis in the proximal tubules isolated from CLP kidneys. These included basal and compensatory glycolysis as well as glycolytic capacity and reserve. Altered glucose metabolism was also observed in a recent clinical study in patients with or without post-operative AKI after cardiopulmonary bypass surgery [69]. Compared with non-AKI patients, those with AKI demonstrated reduced net renal glucose release and an increase in net renal lactate release. This indicates impaired renal glucose production and lactate clearance in AKI. Additionally, in a retrospective cohort, they showed that changes in glucose metabolism had a strong correlation with mortality in critically ill patients.

The emergence of RNA-sequencing (RNAseq) has allowed for transcriptome-wide analyses that identify the biological pathways that are implicated in the pathophysiology of various diseases. Multiple studies have utilized RNAseq to identify novel mechanisms of AKI, including the critical genes involved in kidney disease progression and recovery in the last few years [72,73,74,75,76]. Park et al. used RNAseq to identify the gene expression changes that occur in multiple phases of in human AKI, including pre-ischemia, during ischemia, and during reperfusion [74]. Interestingly, dysregulated metabolism and apoptotic pathways were found in the ischemic phase of injury, while cell growth and innate immune pathways were involved in the reperfusion phase. Single cell RNAseq has taken this technology even further by allowing for transcriptome analyses in specific nuclei. Transcriptomic signals have also been proposed as predictive and prognostic biomarkers that would allow clinicians to determine the patients with progressive AKI, and thus may require closer follow up. In a bench to bedside study, RNAseq analysis was used in a mouse model of folic-acid nephropathy to track the development of fibrosis [72]. The authors then evaluated for downstream proteins in urine samples as novel, noninvasive biomarkers and found higher levels in CKD patients compared with healthy controls.

In a study of experimental ischemic AKI, bulk kidney RNA-seq analysis showed decreased expression of some gluconeogenesis and increased expression of some glycolysis genes at 24 h that persisted for up to one week [69]. Single nucleus RNA sequencing (snRNA-seq) to investigate tubular responses during the injury and recovery phases showed an extensive transcriptomic response in the proximal tubules. In control mice, the expression of rate-limiting enzymes of gluconeogenesis was largely limited to proximal tubules, whereas the expression of glycolytic enzymes was seen in the thick ascending limb at low levels. In the early phase after IR, in addition to the expected altered expression of genes linked with tubular function, cellular injury, survival, and proliferation, the expression of gluconeogenic enzymes was significantly decreased and the expression of glycolytic enzymes was mildly increased in proximal tubules. This was accompanied by downregulation of transcriptional regulators of gluconeogenesis genes. In the late stage after IR (day 28), most proximal tubular cells showed a similar pattern of transcriptional response to that of control mice suggesting recovery, but a small subset of cells displayed a transcriptomic signature consistent with an altered metabolic state.

In parallel experiments to investigate the transcriptional response in clinical AKI, renal biopsies in kidney transplant patients were obtained shortly after reperfusion in brainstem death donors to capture the early response to ischemia and were compared with protocol allograft biopsies obtained 12 months after transplantation to capture the recovery state [69]. RNA-seq analyses showed reduced gluconeogenic and relatively increased glycolytic enzyme mRNA levels in the early phase. In comparison with living donor kidneys with a shorter phase of ischemia, the deceased donor kidneys showed downregulated gluconeogenic genes. In repeat biopsy samples from kidney transplant patients, gluconeogenic enzymes were reduced and glycolytic enzymes were increased in patients with CKD or early transition to CKD, compared with those with normal kidney functions after transplant. They also found that decreased gluconeogenic enzyme expression correlated with decreased kidney function at 1 year. These findings have important implications for novel drug or biomarker development focused on specific phases of kidney injury, as well as specific cell populations. Transcriptomic analysis of critical pathways involved in kidney metabolism is worthy of future investigation.

## 4. Therapeutic Targeting of Mitochondrial Dysfunction and Metabolism in AKI

Therapeutic approaches targeting mitochondria and metabolism have become an active area of investigation given the accumulating evidence supporting their role in the pathophysiology of AKI. Potentially modifiable targets include mitochondrial ROS, biogenesis, mitochondrial fragmentation, metabolites, and metabolic enzymes in energy generation pathways. We discuss below the therapies that have demonstrated promise in pre-clinical models, and some that have been explored in clinical studies as referenced in Table 2.

### 4.1. Mitochondrial Targeted Antioxidants

Various mitochondrial targeted antioxidants have been tested to boost ROS scavenging capacity which is impaired in AKI. Coenzyme Q10 (CoQ10; ubiquinone) has been shown to be renoprotective in pre-clinical models and is in phase III clinical trials for mitochondrial disorders and Parkinson’s disease (*ClinicalTrials.gov*, NCT00432744 (accessed on 30 June 2021)). MitoQ is a derivative of CoQ10 with increased mitochondrial uptake and has been shown to attenuate ischemic and cisplatin induced AKI when administered before injury [77,78] and is currently in clinical trial for CKD (*ClinicalTrials.gov*, NCT02364648 (accessed on 30 June 2021)). Other antioxidants include Mito-CP and SkQR1, which have also shown to be beneficial in ischemic and nephrotoxic AKI [78,79]. Lastly, a mimic of mitochondrial-specific superoxide dismutase, GC4419, was shown to mitigate AKI after cisplatin [80].

### 4.2. Mitochondrial Biogenesis, Function and Quality Control

In pre-clinical models of AKI, various drugs have been tested to stimulate mitochondrial biogenesis and increase the healthy mitochondrial pool. The majority of these drugs improve mitochondrial biogenesis via activation of PGC1α. Two main upstream regulators of PGC1α are AMP-kinase (AMPK) and sirtuins as discussed below. Other pharmacological agents to improve mitochondrial biogenesis include formoterol, a β2 adrenergic receptor agonist, which was shown to be protective in a rat model of IR [38]. A 5-hydroxytryptamine receptor 1f agonist, LY344864, was similarly shown to induce mitochondrial biogenesis and improve renal function in a mouse model of IR [81]. Other mitochondrial quality control targets include inhibiting mitochondrial fragmentation from mitochondrial fission. Drp1 is an important inducer of mitochondrial fission and its pharmacological inhibition by mdivi-1 has been shown to reduce mitochondrial fragmentation, protect against AKI, and promote cellular repair in pre-clinical studies [28]. Another pharmacological agent shown to be protective against ischemic and cisplatin-induced AKI is mitochonic acid 5 which binds to mitofilin in the IMM and facilitates ATP generation [82].

#### 4.2.1. AMP-Kinase (AMPK)

The AMPK signaling pathway impacts various aspects of mitochondrial function and metabolism. AMPK is identified as a principle sensory protein, which monitors the cellular energy status through sensing the ratio of ADP/ATP or AMP/ATP or both. It is activated in response to an increased AMP/ATP ratio, which is induced by hypoxia, ischemia, ROS, or DNA damaging reagents. After activation, AMPK stimulates glucose uptake and fatty acid oxidation for energy generation and decreases processes such as protein and lipid synthesis which consume energy. The AMPK pathway plays a major role in the physiological regulation of mitochondrial homeostasis, including mitochondrial biogenesis, mitophagy, and dynamics [83,84,85]. AMPK activates PGC1α by phosphorylation and stimulates mitochondrial biogenesis. An AMP analog, 5-aminoimidazole-4-carboxamide-1-β-D-riboside (AICAR), activates AMPK by increasing the AMP/ATP ratio. AICAR induced activation of AMPK increased expression of PGC1α, restored mitochondrial content and function, and ameliorated ischemic kidney injury in rats [86]. Furthermore, treatment with AICAR prevented mitochondrial fragmentation and dysfunction, and tubular damage in mice with cisplatin-induced AKI [87].

#### 4.2.2. Sirtuins

Sirtuins (SIRT) are a family of NAD^+^ dependent protein deacetylases and have significant effects on mitochondrial function. Nuclear SIRT 1 acts as a positive transcriptional regulator which enhances the transcription of many mitochondrial-related genes, including PGC1α [88,89] by its deacetylation. Activation of SIRT1 has been shown to stimulate mitochondrial biogenesis and protect against AKI [90,91]. SRT1720, a synthetic activator of SIRT1, promoted renal recovery from ischemia by inducing PGC-1α-regulated mitochondrial biogenesis [92]. SIRT3, present in the mitochondrial matrix, regulates various aspects of mitochondrial function and promotes oxidative phosphorylation. SIRT3 maintains mitochondrial homeostasis through regulation of ETC efficiency [93], mtDNA integrity [94], mitochondrial biogenesis, and fission [87], as well as mitochondrial oxidative stress [93,95]. SIRT3 overexpression has been shown to be protective in sepsis and cisplatin induced AKI and SIRT3 knockout resulted in worse injury in sepsis induced AKI [96,97].

#### 4.2.3. Szeto–Schiller Peptides

Szeto–Schiller Peptides (SS peptides) are newer compounds, which optimize mitochondrial ETC efficiency and restore the cellular bioenergetics [98]. Specifically, SS-31 binds to cardiolipin, an inner mitochondrial membrane phospholipid, which regulates cristae formation and ETC organization [99]. It prevents cardiolipin peroxidation and release of cytochrome C in the cytosol, protects cristae formation, stabilizes mitochondrial structure, and enables efficient electron transport with reduced ROS generation. It has been shown to recover ATP production, cristae formation, and improve renal function in IRI-induced AKI [45,99,100]. SS-31, also known as MTP-131 and elamipretide, is currently in multiple clinical trials, including those for treating impaired kidney function (*Clinicaltrails.gov*. NCT02436447 (accessed on 30 June 2021)).

### 4.3. Pharmacological Strategies Targeting Metabolism

As discussed above, the PDH-PDK axis acts as a gatekeeper for energy metabolism. Activation of PDK selectively inhibits PDH, which is an enzyme that converts pyruvate to acetyl-CoA and results in pyruvate being converted to lactate. Recently, inhibition of PDK activation using pharmacologic and genetic approaches was shown to attenuate cisplatin-induced acute kidney injury [101]. Similarly, a protective role of the PDK inhibitor, dichloroacetate, in cisplatin-induced acute kidney injury has been demonstrated [102]. Other strategies have included supplementation with various substrates of TCA cycle including α-ketoglutarate + aspartate, malate, succinate, citrate, and fumarate during reoxygenation after anoxia in isolated proximal tubules [103,104]. In the clinical scenario, high-dose thiamine supplementation in the first 4 h after ICU admission in AKI patients showed a rapid decrease in impaired metabolism and clearance of lactate along with lowered mortality [69]. Thiamine functions as a co-factor (in the form of thiamin pyrophosphate, TPP) in oxidative energy metabolism and ATP production within the mitochondria. Another substrate that has been recently investigated is NAD^+^, which is an essential cofactor and substrate for metabolic enzymes in energy generation within the mitochondria and for mitochondrial sirtuins [105]. Its reduced form, NADH, carries high-energy electrons from oxidation of various substrates in energy metabolism including glycolysis, FAO, and TCA cycle from oxidation of various substrates in energy metabolism. Its de novo biosynthesis was recently discovered as a downstream effector to PGC-1α mediated protection in AKI [41]. PGC-1α knockout mice exhibit high sensitivity to ischemia, and NAD^+^ augmentation in the form of its precursor nicotinamide (NAM) diminished the response to ischemic-stress similar to controls [41]. Other studies have shown similar renoprotective effects of NAD^+^ supplementation in the form of another precursor, nicotinamide mononucleotide, in cisplatin induced AKI in aged mice [106]. Boosting de-novo NAD^+^ biosynthesis by dietary supplementation with TES-1025 improved mitochondrial metabolism via enhanced fatty acid oxidation, antioxidant capacity, and kidney function in cisplatin induced AKI [107]. Importantly, pre-operative NAM administration has been shown to be beneficial with lower rates of AKI in a pilot randomized clinical trial in patients undergoing cardiac surgery [108]. All the studies reviewed above indicate that pharmacological interventions targeting mitochondrial function and metabolism could serve as favorable strategies to mitigate AKI.

## 5. Summary and Future Perspectives

Increasing evidence supports the disruption of mitochondrial homeostasis and metabolism as a critical component in the pathogenesis of AKI. Alterations in mitochondrial morphology and function represent early events during AKI, which appear to precede the initiation of tubular dysfunction and cell death. Mitochondrial injury not only contributes to the development of AKI, but also may impact the recovery from AKI. Changes in tubular metabolism also appear to play an important role in AKI pathophysiology. Numerous pharmacological modulators targeting mitochondria and metabolism have been investigated and are already in use in other fields. However, there are several unanswered questions and limited understanding of some fundamental aspects which are pertinent to the success of potential therapeutics.

Mitochondrial dysfunction and metabolic reprogramming are often found together under stress conditions. Whether early changes mitochondrial function/morphology are primary (*causative*) in tubular injury or secondary (*consequence*) to changes in energy demand or other primary events is not clear. This may also differ between various etiologies of AKI. For instance, cisplatin directly targets mitochondrial function, by accumulating in mitochondria and suppressing OXPHOS by decreasing the activity of ETC complexes. Mechanisms involved during cell injury may be largely focused on survival and cellular hibernation. This may result in an adaptive decline in mitochondrial respiration, since ATP demand is its major driver. With reduced GFR and filtered load, tubular mitochondrial respiration may decrease due to lower ATP demand. Administering therapeutics at this stage to increase mitochondrial respiration may not provide much benefit.

Another related aspect is the structural-functional relationship in mitochondria. As discussed above, altered mitochondrial morphology and dynamics have been uniformly seen in different etiologies of AKI. Mitochondrial dynamics and morphology also regulate bioenergetic capacity and efficiency. Whether the morphological changes drive the bioenergetic changes in AKI or vice versa is not known. Therapeutics targeting increased substrate utilization and OXPHOS are less likely to be successful in situations with damaged mitochondrial morphology. In this regard, drugs targeting mitochondrial biogenesis are likely to be most effective on newly generated mitochondria with intact morphology and function.

The impact of mitochondrial dysfunction and metabolic reprogramming may differ during injury from recovery and in the repair phases. During repair and recovery, biosynthetic mechanisms to support cellular proliferation are likely to predominate. An early glycolytic phase post-injury with subsequent transition to OXPHOS would likely restore both structural and functional recovery. The role of metabolic reprogramming as an early, adaptive response or a persistent, maladaptive response and the impact on progression of injury vs. repair and recovery needs to be studied in further details. A better understanding of this will inform the timing of therapeutics targeting metabolism to increase the likelihood of their success. Relevant to this, most investigational therapies are focused on prevention of AKI, which could be successful in the circumstances where the timing of injury is known and predictable such as post-cardiac surgery and contrast nephropathy. However, frequently in a clinical setting, the timing of injury is not known. Therefore, therapies focused on repair and recovery in AKI are likely to be more beneficial. This strategy may also be successful across various etiologies of AKI. While mechanisms of injury may differ between types of AKI, the mechanisms of cellular proliferation, and redifferentiation for repair and recovery are the same regardless of the etiology of AKI. A more complete understanding of the molecular mechanism by which mitochondria respond to injury and recovery, as well as the upstream regulators and downstream effects in kidney disease, could facilitate the development of additional therapies of AKI.

## Figures and Tables

**Figure 1 jcm-10-03991-f001:**
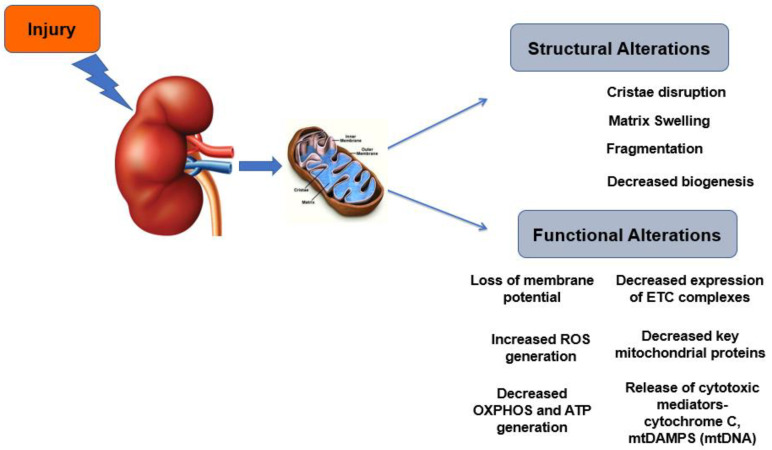
Mitochondrial structural and functional alterations in AKI.

**Table 1 jcm-10-03991-t001:** Acute kidney injury (AKI) definition and staging.

**AKI Definition**	**Serum Creatinine or GFR Criteria**	**Urine Output Criteria**
	Increase in serum creatinine by ≥0.3 mg/dL (≥26.5 micromol/L) within 48 h, **or** Increase in serum creatinine to ≥1.5 times baseline, which is known or presumed to have occurred within the prior seven days, **or**	Urine output <0.5 mL/kg/hour for 6 h
**AKI Staging**	**Serum Creatinine or GFR Criteria**	**Urine Output Criteria**
**Stage 1**	Increase in serum creatinine to 1.5 to 1.9 times baseline, **or** Increase in serum creatinine by ≥0.3 mg/dL (≥26.5 micromol/L), **or**	Reduction in urine output to <0.5 mL/kg/hour for 6 to 12 h.
**Stage 2**	Increase in serum creatinine to 2.0 to 2.9 times baseline, or	Reduction in urine output to <0.5 mL/kg/hour for ≥12 h.
**Stage 3**	Increase in serum creatinine to 3.0 times baseline, or Increase in serum creatinine to ≥4.0 mg/dL (≥353.6 micromol/L), **or** Initiation of kidney replacement therapy, **or** In patients < 18 years, decrease in estimated glomerular filtration rate (eGFR) to <35 mL/min/1.73 m^2^.	Reduction in urine output to <0.3 mL/kg/hour for ≥24 h, or Anuria for ≥12 h

GFR: Glomerular filtration rate; in bold or: Any one of these criteria meet the definition of AKI.

**Table 2 jcm-10-03991-t002:** Clinical trials with drugs impacting mitochondrial function and metabolism.

Therapy	Mechanism	Clinical Trial	Details
Elamipretide (MTP-131)	Prevents the peroxidation of cardiolipin by cytochrome c	Phase I NCT02436447The Safety and Pharmacokinetics of Repeat-dose Intravenous Infusion of MTP-131 in Subjects with Impaired Renal Function (2015)	Open-label, parallel group, multiple dose study to evaluate the safety, tolerability, and pharmacokinetics of one-hour intravenous infusion of MTP-131 administered for 7 consecutive days.
ASP1128	Selective PPARδ modulator, promotes fatty acid oxidation	Phase II NCT03941483Evaluate the Efficacy of ASP1128 (MA-0217) in Subjects at Risk for Acute Kidney Injury Following Coronary Artery Bypass Graft (CABG) and/or Valve Surgery (2019)	Double-blind study to investigate the safety and tolerability of postsurgery treatment with ASP1128, and pharmacokinetic characteristics of ASP1128 in subjects at risk for AKI following CABG and/or valve surgery.
Nicotinamide	Incorporates into nicotinamide adeninedinucleotide (NAD)^+^ and NADP^+^,coenzymes in enzymaticoxidation-reduction reactions.	Phase II NTC04342975Evaluate the Efficacy of BASIS™ (Nicotinamide Riboside and Pterostilbene) Treatment for Kidney Protection in Patients Treated by Complex Aortic Aneurysm Repair and Aortic Arch Reconstruction (2020)	Single-center, prospective, randomized, double-blinded, placebo-controlled phase II clinical trial to evaluate the efficacy of “NAD^+^ supplementation” in preventing AKI in patients undergoing complex aortic aneurysm repair and open aortic arch reconstruction.
MitoQ	Antioxidant, derivative of CoQ10 with increased mitochondrial uptake	Phase IV NCT02364648Mitochondrial Oxidative Stress and Vascular Health in Chronic Kidney Disease (2017)	Controlled, double blinded trial, Stage 3–5 chronic kidney disease (CKD) patients will be randomly assigned to receive a 4-week daily dose of a mitochondria targeted antioxidant (MitoQ) or a placebo.

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
