# Peer review of "Targeting Mitochondria and Metabolism in Acute Kidney Injury"

_jcm, 2021, doi:10.3390/jcm10173991_

Round 1

Reviewer 1 Report

The text is complete regarding the description of mitochondrial damage mechanisms such as altered biogenesis and functional alterations. Finally, the preclinical and clinical studies that can act on mitochondrial alterations are listed. The topics have been dealt with in other review papers doi: 10.1097 / MNH.0000000000000228 and doi: 10.1016 / j.semnephrol.2020.01.002
The text may undoubtedly be of interest to biologist and biotechnologist readers. However, I think the manuscript can be improved to make it interesting for a wider audience.

Major:

1) The text is very discursive; although rich in detail, the use of diagrams, figures and tables could make it easier to understand. For example, I would suggest adding a table with ongoing clinical trials, a scheme of metabolic pathways and therapeutic agents.

2) A definition of Acute kidney injury (AKI) is missing, and the authors treat the topic of Acute kidney injury (AKI) in a very general way: AKI has several etiologies (doi: 10.1016 / S0140-6736 (19) 32563-2.). The damage mechanisms for each syndrome are sometimes different. To make it different from other review papers that have already dealt with the topic, I would suggest arguing and create schemes the mitochondrial damage in the different types of AKI.

3) Summarize in one paragraph the current therapies for AKI if and how they affect mitochondrial metabolism.

Minor:

1) add a list of the abbreviations mentioned in the text, please.

Author Response

The text is complete regarding the description of mitochondrial damage mechanisms such as altered biogenesis and functional alterations. Finally, the preclinical and clinical studies that can act on mitochondrial alterations are listed. The topics have been dealt with in other review papers doi: 10.1097 / MNH.0000000000000228 and doi: 10.1016 / j.semnephrol.2020.01.002. The text may undoubtedly be of interest to biologist and biotechnologist readers. However, I think the manuscript can be improved to make it interesting for a wider audience.

  • We appreciate the constructive critiques.

Major:

1) The text is very discursive; although rich in detail, the use of diagrams, figures and tables could make it easier to understand. For example, I would suggest adding a table with ongoing clinical trials, a scheme of metabolic pathways and therapeutic agents.

  • As suggested, we have included figures and tables for better clarity.

2) A definition of Acute kidney injury (AKI) is missing, and the authors treat the topic of Acute kidney injury (AKI) in a very general way: AKI has several etiologies (doi: 10.1016 / S0140-6736 (19) 32563-2.).

  • We have included the definition and staging of AKI along with additional details to address this concern.

The damage mechanisms for each syndrome are sometimes different. To make it different from other review papers that have already dealt with the topic, I would suggest arguing and create schemes the mitochondrial damage in the different types of AKI.

  • We have focused on the published findings and discussion of the three most common etiologies of AKI including ischemia, nephrotoxic and sepsis induced kidney injury and have indicated which mechanisms have been observed in the different etiologies of AKI.

3) Summarize in one paragraph the current therapies for AKI if and how they affect mitochondrial metabolism.

  • Currently, there are no approved or specific therapies for AKI. We have provided the details of potential therapies which impact mitochondrial metabolism (Table 2) and Summary and Future Perspective section to address this.

Minor:

1) add a list of the abbreviations mentioned in the text, please.

  • We have added this.

Reviewer 2 Report

The topic is relevant to an important question and this review is useful considering the recent advances  in the field.

The manuscript is organised in 3 parts: physiology, pathophysiology, and therapeutic perspectives.  The first 2 parts are comprehensive and well focused.

In the 3rd part, promising therapeutic strategies are described. It would be interesting if the authors went beyond describing existing data, and provided their personnel hypotheses and conviction, and identify the main milestones and pitfalls on the road to a clinical implementation of mitochondria-oriented treatments in AKI.

Overall, the is a good review, with a clear interest for the readership of JCM.

Author Response

The topic is relevant to an important question and this review is useful considering the recent advances in the field. The manuscript is organised in 3 parts: physiology, pathophysiology, and therapeutic perspectives.  The first 2 parts are comprehensive and well-focused.

  • We appreciate the kind and encouraging comments.

In the 3rd part, promising therapeutic strategies are described. It would be interesting if the authors went beyond describing existing data, and provided their personnel hypotheses and conviction, and identify the main milestones and pitfalls on the road to a clinical implementation of mitochondria-oriented treatments in AKI.

We have significantly expanded the Summary and Future Perspective section to address this.

Round 2

Reviewer 1 Report

The work has now been substantially improved. The addition of the AKI definitions allows for easy reading to a non-specialized audience. The update on clinical studies is interesting, as well as the image describing the pathophysiology of AKI. I congratulate the authors and suggest that this work be published.